# Fabrication of TiO_2_ – KH550 – PEG Super-Hydrophilic Coating on Glass Surface without UV/Plasma Treatment for Self-Cleaning and Anti-Fogging Applications

**DOI:** 10.3390/ma15093292

**Published:** 2022-05-04

**Authors:** Muhammad Nobi Hossain, Sung-Jun Lee, Chang-Lae Kim

**Affiliations:** 1BK21 FOUR Program for Development of Regional Future Engineers in Smart Mechanical Components, Chosun University, Gwangju 61452, Korea; mnhossain.8911@gmail.com (M.N.H.); k3668591@chosun.kr (S.-J.L.); 2Department of Mechanical Engineering, Chosun University, Gwangju 61452, Korea

**Keywords:** TiO_2_, TiO_2_ – KH550 – PEG composite film, anti-fogging, self-cleaning, super-hydrophilicity

## Abstract

In this study, we have developed a self-cleaning transparent coating on a glass substrate by dip coating a TiO_2_ – KH550 – PEG mixed solution with super-hydrophilicity and good antifogging properties. The fabrication of the thin-film-coated glass is a one-step solution blending method that is performed by depositing only one layer of modified TiO_2_ nanoparticles at room temperature. X-ray diffraction (XRD) and scanning electron microscopy (SEM) were used to determine the structure and morphology of the nanoparticles and the thin-film-coated glass. The surface functional groups were investigated using Fourier-transform infrared spectroscopy (FT-IR), and the optical properties of the glass coating were measured using a UV/Vis spectrometer. The results revealed that the KH-500-modified TiO_2_ film coating was in an anatase crystalline form. The hydrophilicity of the coated and uncoated glass substrates was observed by measuring their water contact angle (WCA) using a contact angle instrument. The maximum transparency of the coated glass measured in the visible region (380–780 nm) was approximately 70%, and it possessed excellent super-hydrophilic properties (WCA ~0°) at an annealing temperature of 350 °C without further need of UV or plasma treatment. These results demonstrate the super-hydrophilic coated glass surface has potential for use in self-cleaning and anti-fogging applications.

## 1. Introduction

Our living environment is surrounded by various hazardous and toxic pollutants such as volatile organic compounds (VOCs), polycyclic aromatic hydrocarbon (PCAHC), heavy metals such as Pb, Cd, Cu, Cr, Ni, and Se, and dust particles (PM2.5–PM10). The presence of these pollutants in the environment not only affects human life, but also damages and shortens the life of buildings and construction materials. Pollutants in the air are easily deposited on the surface of building materials (glass, ceramics, steel, and roofing tiles) via physiochemical reactions, which causes the degradation of building materials and decorations [1,2,3,4].

At present, the surfaces of building materials used for outdoor applications are exposed to high levels of pollution. In addition, the deposition of dust impurities on solar panel cover glass reduces both the transparency of the solar panel and the efficiency of its electrical performance [5,6,7,8]. Maintaining the cleanliness of the surfaces of building materials, window glasses, and solar panels via traditional cleaning methods such as sanitizing or other detergents not only leads to the release of toxic chemicals into the environment but also increases economic burden [9,10]. An efficient way to protect the surfaces of construction/building materials is to coat the surfaces with photoactive self-cleaning materials. Recently, self-cleaning materials have drawn considerable attention worldwide owing to their unique properties and wide range of applications in energy and environmental areas, such as window glasses, car mirrors, textiles, fabrics, and solar panels [11,12,13,14].

Extensive research is being conducted to develop highly efficient self-cleaning surfaces with high durability. Self-cleaning surfaces can be classified into two categories: super-hydrophobic (WCA > 150°) and super-hydrophilic (WCA < 10°). A well-known example of a super-hydrophobic surface is the lotus leaf. In the case of super-hydrophobic self-cleaning surfaces, water droplets slide and roll over the surface, thereby cleaning it. In super-hydrophilic coated surfaces, water droplets spread over the surface and wash away dust particles and other contaminants [15,16,17,18,19]. In this study, we focused on developing a super-hydrophilic surface coating to save cleaning time and reduce maintenance costs.

Various metal oxides such as TiO_2_, WO_3_, ZnO, and Fe_3_O_4_ have been investigated for developing self-cleaning surfaces [20,21,22,23]. To date, self-cleaning coatings based on TiO_2_ have attracted considerable attention for various energy and environmental applications. TiO_2_ is highly stable, both chemically and physically, and it exhibits excellent photoelectrochemical properties, mechanical strength, photocatalytic ability, hydrophilic properties, good transparency, protection from ultraviolet, environmental friendliness, and low cost. Several studies have been conducted on the fabrication of self-cleaning TiO_2_ coatings on glass substrates [24,25,26,27,28,29]. However, in some cases, UV light irradiation and plasma treatment are required to obtain a stable super-hydrophilic wetting surface, which is time consuming and also increases the manufacturing cost [30,31,32,33]. There are few works available regarding the fabrication of super-hydrophilic self-cleaning TiO_2_-coated surfaces that do not require any additional treatments (plasma/UV). Therefore, it is desirable to develop a transparent and super-hydrophilic TiO_2_ film without the need for plasma or UV light illumination. In this study, we successfully developed a stable, super-hydrophilic, single-layer coating on a glass substrate without the need for UV light irradiation or plasma treatment, using KH550-modified TiO_2_ nanoparticles by a simple dip coating technique.

Generally, inorganic nanoparticles tend to agglomerate easily in the reaction medium and their dispersion capability in organic solvents is poor, which greatly limits the application of TiO_2_ nanoparticles. Surface modification of TiO_2_ nanoparticles is an effective method of improving their dispersibility and reduce agglomeration in the medium by decreasing the surface energy [34,35]. Silane coupling agents are commonly used in the surface modification of TiO_2_ nanoparticles because of their simplicity. In addition, silica is widely employed as a super-hydrophilic agent with excellent transparency and anti-fogging properties, owing to its high surface concentration of hydroxyl (OH-) groups. The formation of a Ti-O-Si covalent bridge between nanoparticles and silane agents in the surface modification medium plays a role in functionalizing the surface of TiO_2_ nanoparticles [36,37].

In this study, (3-aminopropyl) triethoxysilane (KH550) and tetra-ethyl-orthosilicate (TEOS) were selected as the silica sources used to modify the surface of TiO_2_ nanoparticles, polyethylene glycol (PEG) was used as the surfactant and binder, and ethanol was used as the organic solvent. Finally, a super-hydrophilic self-cleaning TiO_2_ – KH550 – PEG composite coating was developed on a glass substrate using a dip coating technique. The presence of a hydroxyl group on the surface of TiO_2_ nanoparticles played a critical role in the hydrophilicity of the TiO_2_ – KH550 – PEG -coated glass. Surface modification of TiO_2_ nanoparticles with a KH550 silane coupling agent is presented in Figure 1. The characteristics, properties, morphology, and performance of the TiO_2_ nanoparticles before and after modification were investigated. The TiO_2_ super-hydrophilic coatings demonstrated excellent self-cleaning and anti-fogging properties without the assistance of UV/plasma treatment. While there is a tendency of silica-modified TiO_2_ film coatings to lose hydrophilicity in dark environments because of the replacement of hydroxyl groups by oxygen from the air [38], the prepared TiO_2_ – KH550 – PEG /glass substrate exhibited stable super-hydrophilicity in a dark environment.

## 2. Experimental Setup

### 2.1. Materials

For preparing TiO_2_ – KH550 – PEG sol, the following chemicals were used: Titanium (IV) dioxide (TiO_2_) Nanopowder (≥99.5%, Sigma Aldrich); Polyethylene glycol (PEG, M_av_ = 400 H(OCH_2_CH_2_)_n_ OH), (Daejung Chemicals & Metals, South Korea); (3-aminopropyl) triethoxysilane H_2_N(CH_2_)_3_Si(OC_2_H_5_)_3_, KH550), (98%, Sigma Aldrich); Tetraethyl orthosilicate (TEOS), Si(OC_2_H_5_)_4_ (≥99.0%, Sigma Aldrich, USA); Ethyl alcohol (C_2_H_5_OH) (96%, Samchun Pure Chemicals, South Korea); and Deionized water.

### 2.2. Preparation of TiO_2_ Hydrophilic Film (TiO_2_ – KH550 – PEG ) and Deposition on Glass Substrate

#### 2.2.1. Preparation of TiO_2_ – KH550 – PEG Hydrophilic Film

A modified TiO_2_ porous film was prepared at room temperature in the following manner: a certain amount of PEG was added to a 30 mL solution of ethanol and was stirred until completely dissolved. PEG was used as the binding and structure-directing agent. Subsequently, 3 g of TiO_2_ and the required amount of water were added to the mixture. The reaction was performed under vigorous stirring. After stirring for 1 h, an appropriate amount of surface modifier (KH550) and silane coupling agent (TEOS) were added dropwise into the above-mentioned reaction mixture and stirred for 2 h. The mixed TiO_2_ sol was ultrasonicated for 30 min to ensure good dispersion of TiO_2_ nanoparticles. In this process, silane groups were deposited onto the surface of the TiO_2_ nanoparticles via condensation, and bonds were established between the silanol groups and nanoparticles. Pure TiO_2_ and TiO_2_ – KH550films were also prepared for comparison purposes by dip coating using each component solution.

#### 2.2.2. Deposition of TiO_2_-KH550-PEG Thin Film on Glass Substrate

##### Pretreatment

Soda-lime glass slides (Microscope glass slide, Paul Marienfeld GmbH & Co.KG, Lauda-Königshofen, Germany) with dimensions of 76 mm × 26 mm × 1 mm were used as substrates and were pretreated before TiO_2_-based films were deposited. The glass substrates were cleaned at first with detergent; thereafter, by sonicating in an ethanol solution, acetone solution and ionized (DI) water, respectively, for more than 10 min. To remove moisture from the surface, all the glass slides were dried in an oven at 100 °C for 10 min, cooled down to room temperature, and stored for dip coating.

##### Deposition

TiO_2_ films were deposited onto the pre-cleaned glass slides using a dip coating method in a TiO_2_ – KH550 – PEG solution with an immersion rate of 20 mm/min for 20 s and withdrawal rate of 20 mm/min. After 2 min of immersion, each coated glass substrate was removed, air-dried for 10 min, and oven dried at 150 °C for 12 h. To remove unreacted polymers and complete condensation, TiO_2_ – KH550 – PEG films were annealed at 350 °C for 1 h, and thereafter, stored for characterization and future applications. The thickness of the TiO_2_-KH550-PEG film measured by a 2D profilometer (SV2100, Mitutoyo, Kawasaki, Japan) was approximately 330 nm.

### 2.3. Characterization of TiO_2_-KH550-PEG Thin-Film-Coated Glass Surface

#### 2.3.1. X-ray Diffraction Analysis (XRD)

The crystallinity of the KH550-modified TiO_2_-coated glass was measured by a high-resolution X-Ray diffractometer (XRD) (X’Pert PRO multi-purpose X-ray diffractometer) at an accelerating voltage of 40 kV and a current of 40 mA with a scanning rate of 5°, 2 θ min ^−1^.

#### 2.3.2. FT-IR Spectroscopy

The FT-IR spectra of the TiO_2_ film coating were recorded on a spectrum 400 in wavenumbers ranging from 400–4000 cm ^−1^ to characterize the surface groups of the modified-TiO_2_-coated and uncoated glass substrates.

#### 2.3.3. Field Emission Scanning Electron Microscope (FESEM)

The FESEM images of nanocoated and uncoated glass were obtained using a SEM model Gemini 500 + EDS (Oxford) to observe the morphology of the TiO_2_ film.

#### 2.3.4. Water Contact Angle (WCA) Analysis and Optical Properties Measurement

The WCA of the coated glass was quantified by measuring the water droplet particle size on the surface of the coated glass using a microscope camera (U1000X). The optical properties (light transmittance) of the coated glass were measured by UV-VIS spectrometer (OPTIZEN POP, K LAB Co., LTD., Daejeon, South Korea).

#### 2.3.5. Self-Cleaning Property Analysis and Adhesion Test

The self-cleaning performance of the highly transparent super-hydrophilic surface was investigated using a colored metal solution as a dirt contaminant. The prepared colored metal solution was dropped onto bare and coated glass using a micropipette. Subsequently, the self-cleaning properties of the TiO_2_-coated and bare glass surfaces were investigated. The adhesive properties of the coated specimens were evaluated using a peeling test.

#### 2.3.6. Anti-Fog Property Analysis

The anti-fog activity of the coated and uncoated glass was examined by placing them in a boiling-hot water bath for 10 min. Then, the development of haze and tiny water droplets on the coated and uncoated glass was investigated. Subsequently, both coated and bare glass slides were placed on paper to observe the anti-fog properties of the glass surfaces.

## 3. Results and Discussion

### 3.1. XRD Analysis

Figure 2 depicts the X-ray diffraction pattern of pure TiO_2_ and modified TiO_2_ film coatings. From the figure, we can see that there is one sharp peak at 2θ = 25.41 in both pure and KH550-modified film, corresponding to the anatase crystalline structure of nano-TiO_2_ [39,40,41]. Therefore, it can be concluded that the addition of KH550 and PEG did not alter the crystalline structure of the nano-TiO_2_.

### 3.2. FT-IR Analysis

The pure TiO_2_ film and modified TiO_2_ – KH550 – PEG composite film coated on a glass substrate were characterized using FT-IR spectroscopy (Figure 3). It can be seen from Figure 3 that new peaks appeared in the analysis of the modified TiO_2_ – KH550 – PEG composite film. The peaks around 2924 cm ^−1^ and 1400 cm ^−1^ correspond to the stretching vibration of (CH_2_-) and C-H bending which are indicators of the interaction of PEG (or KH550 or residues of TEOS) with the surface of TiO_2_ nanoparticles [42,43,44,45,46]. The peak at around 1700 cm ^−1^ is assigned to the terminal amino group (-NH_2_ or –CH_2_-NH_2_) of silane agent. Another small peak at approximately 1100 cm ^−1^ is assigned to the absorption band of the Si-O-Si bond which also indicates the introduction of the silane agent to the TiO_2_ nanoparticles [35,37,47,48]. These FTIR results suggest that TiO_2_ nanoparticles were successfully modified by KH550 and PEG.

### 3.3. Surface Morphology Analysis

Figure 4 depicts micrographs of pure TiO_2_ film, TiO_2_ – KH550 film undoped with PEG, and TiO_2_ – KH550 – PEG composite film on a glass substrate. It can be seen that, in the pure TiO_2_ film (non-modified with KH550), the nanoparticles were agglomerated (Figure 4a). In the TiO_2_ – KH550 film undoped with PEG, the aggregation of nanoparticles decreased, but many cracks occurred on the surface (Figure 4b). In contrast, we can see that the microstructure of the TiO_2_ – KH550 – PEG film is smooth and uniform, which implies that the addition of PEG as a surfactant and binder improved the surface morphology of the film (Figure 4c). This indicates that, after surface modification with silane agent KH550 and PEG, nano-TiO_2_ can be dispersed more homogeneously on the glass substrate.

### 3.4. Wettability and Super-Hydrophilicity Property of the Coating

The wettability properties of the bare and coated glasses were observed and compared by measuring their static WCA (Figure 5). A water droplet with a volume of 10 µL was dispensed onto bare and coated glass substrates. Images of the water droplets were recorded to measure WCA. As depicted in Figure 5, the bare glass exhibited a WCA of 45°, which is hydrophilic in nature. Pure TiO_2_, TiO_2_ – KH550, and TiO_2_ – KH550 – PEG films all showed WCA close to zero as water droplets spread over the surface. Since TiO_2_ itself has super-hydrophilicity, all films containing TiO_2_ showed super-hydrophilicity. That is, even if KH550 (added to reduce the aggregation of nanoparticles in pure TiO_2_ film) and PEG (added to solve the problem of cracks on the surface of TiO_2_ – KH550 film) were mixed with TiO_2_, individually or simultaneously, their super-hydrophilicity was confirmed to be maintained.

Although the pure TiO_2_ and TiO_2_ – KH550films showed super-hydrophilicity, as a result of the peeling test performed with adhesive tape (Figure 6a), the pure TiO_2_ and TiO_2_-KH550 films on the glass were peeled off (Figure 6b,c), confirming that their adhesion durability was very weak. From the fact that the surface of the TiO_2_ – KH550 – PEG film was not severely damaged (Figure 6d), it can be seen that the adhesion durability of the TiO_2_ – KH550 – PEG film is superior to that of pure TiO_2_ and TiO_2_ – PEG films.

### 3.5. Self-Cleaning Test

A self-cleaning test was conducted in a simple and economical way, in which a metal solution in water was used as a model contaminant. Self-cleaning performance was assessed after each water droplet was dropped (using a micropipette, 10 µL) onto a polluted TiO_2_ – KH550 – PEG composite-coated glass substrate, and a bare glass substrate was prepared for comparison. The schematic of the self-cleaning mechanism of the super-hydrophilic glass substrate is depicted in Figure 7a. From Figure 7b, we can see that the dirt solution was spread out on the surface owing to its super-hydrophilic nature. The cleaning process was investigated by observing the capability of water to remove the dirt solution by the sliding action of water. The dirt solution on the coated glass was completely removed as quickly as the water droplets spread off, resulting in a clean surface. In contrast, complete removal of the dirt solution on the uncoated surface was difficult with the same action, and therefore, a dirt layer was observed on the uncoated glass (Figure 7c). After drying, some solid dirt contaminants were observed on the uncoated glass, whereas no solid contaminants were observed on the coated glass surface. KH550-modified coated glass surfaces exhibited excellent self-cleaning properties with the assistance of water.

After the self-cleaning test, the coated glass was dried in oven at 80 °C for 10 min, and we measured its WCA to observe the changes in the hydrophilicity properties of coated glass. It was observed that super-hydrophilicity (WCA ~0°) was retained after the self-cleaning test, which implies the stability of the super-hydrophilicity of the TiO_2_ – KH550 – PEG -coated glass substrate.

### 3.6. Anti-Fogging Behavior of Coating and Stability of the Hydrophilicity in the Darkness

The anti-fog activity of the bare and coated glass was determined by placing the glass substrates in a boiling water bath for 5 min. Figure 8 depicts the fog development on bare and coated glasses. The results revealed that large droplets appeared on the bare glass (Figure 8a), whereas droplets disappeared on the TiO_2_ – KH550 – PEG -coated glass after the condensation process because of its good wettability (Figure 8b). The super-hydrophilic surface could resist fog formation because water droplets spread on the hydrophilic surface as a thin membrane instead of droplets. The water droplets on the coated glass disappeared completely within 5 min at room temperature. In contrast, the water droplets persisted on the bare glass for 25 min. It can be concluded that the anti-fogging properties of the glass substrate increased with their increasing hydrophilic properties.

There is a possibility of reducing the hydrophilicity of the silica-coated TiO_2_ glass due to the replacement of hydroxyl groups by oxygen from the air. Therefore, the stability of the hydrophilicity of the TiO_2_ – KH550 – PEG -coated glass in a dark environment was tested. The coated glass was kept in a dark environment for a week and WCA was measured in every 24 h (Figure 9). From Figure 9, it can be seen that TiO_2_ – KH550 – PEG -coated glass maintained its super-hydrophilicity over the week, which indicates stable super-hydrophilicity in darkness.

### 3.7. Transmittance Characteristics

To evaluate the transparency of the modified TiO_2_ film, UV-Vis spectra were recorded. The modified TiO_2_ coating exhibited good optical transparency in the visible-near IR range. The maximum light transmittance was 82% for the TiO_2_ – KH550 – PEG -coated glass in the VIS-NIR range (380–1100 nm), whereas it was 92.0% for the bare glass (Figure 10). The transmittance of the TiO_2_ – KH550 – PEG -coated glass measured in the visible range (380–780 nm) was 5–70%. Moreover, in the ultraviolet region (10–400 nm) the TiO_2_ – KH550 – PEG film almost blocked all light, so it can be said that the UV protection performance was excellent. This implies that the TiO_2_ – KH550 – PEG film could be effectively applied for self-cleaning and anti-fogging glass applications.

## 4. Conclusions

A transparent and stable super-hydrophilic coating of a TiO_2_ – KH550 – PEG composite film on a glass substrate was successfully developed by using a simple dip coating technique and an easy preparation process for self-cleaning applications. The formation of a smooth, crack-free, and uniform deposition of the modified TiO_2_ film on the glass substrate was confirmed by XRD, SEM, and FT-IR analyses. It was found that the addition of the KH550 silane agent to the TiO_2_ surface could prevent the agglomeration of TiO_2_ nanoparticles and improve its adhesion to the glass substrate. The developed TiO_2_ – KH550 – PEG /glass substrate exhibited excellent super-hydrophilicity (WCA ~0°) at 350 °C annealing temperature without further treatment by plasma or UV illumination. The WCA of the coated glass was maintained at 0° after exposure to a dark environment for five days, which implies that the surface coating had good durability in a dark environment. The results of self-cleaning analysis revealed that the prepared super-hydrophilic glass had good self-cleaning properties which could be employed to clean contaminants on its surface quickly compared to uncoated glass. The prepared coated glass also exhibited excellent anti-fogging behavior. It was observed that no water droplets formed on the super-hydrophilic surfaces; the thin film of water disappeared within 5 min at room temperature and the glass also maintained its transparency, whereas the bare glass was covered by water droplets and dried completely only after being left for 25 min at room temperature. The maximum transparency of the TiO_2_ – KH550 – PEG -coated glass in the visible range (380–780) was approximately 70%. The developed coating was excellent in UV protection to the extent that it almost blocked light in the ultraviolet region. Therefore, it can be concluded that the prepared TiO_2_ – KH550 – PEG has good potential for applications in windows and architectural glass because of its good transparency, self-cleaning, anti-fogging properties, and adhesion durability. Therefore, further research on transparency is recommended.

## Figures and Tables

**Figure 1 materials-15-03292-f001:**
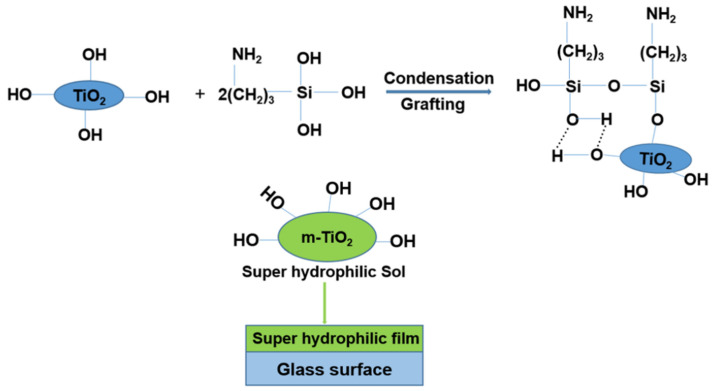
Surface modification of TiO_2_ with (3-aminopropyl) triethoxysilane (KH550) and the modified TiO_2_ (m-TiO_2_) super-hydrophilic film on a glass substrate.

**Figure 2 materials-15-03292-f002:**
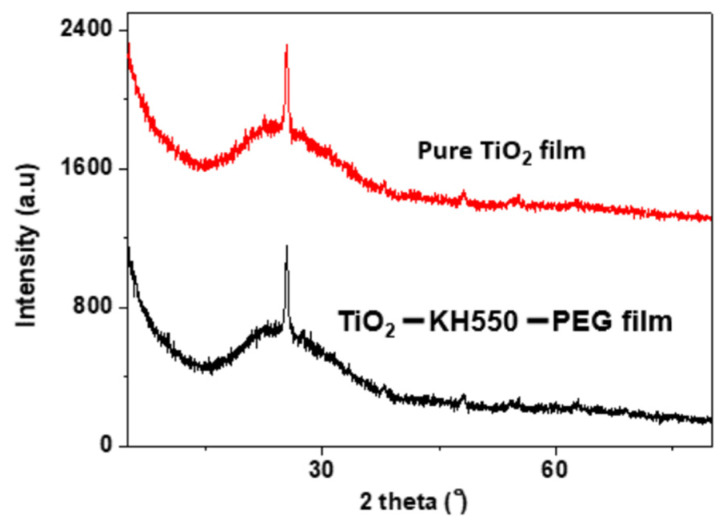
XRD pattern of pure TiO_2_ film and modified TiO_2_ – KH550 – PEG composite film.

**Figure 3 materials-15-03292-f003:**
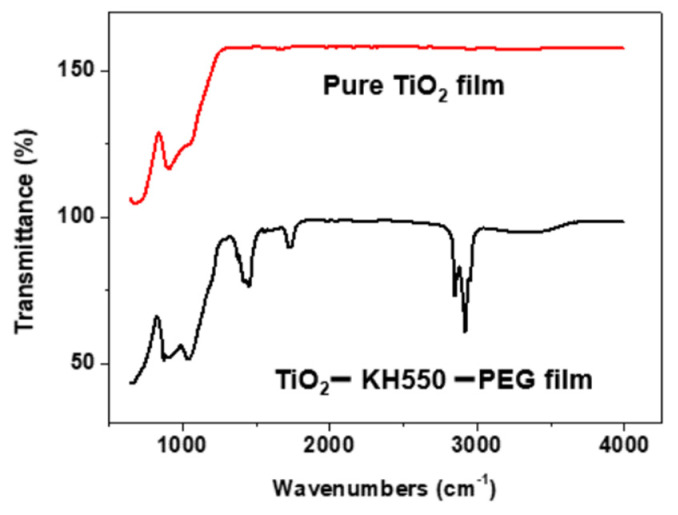
FT-IR spectra of pure TiO_2_ film and TiO_2_ – KH550 – PEG composite film.

**Figure 4 materials-15-03292-f004:**
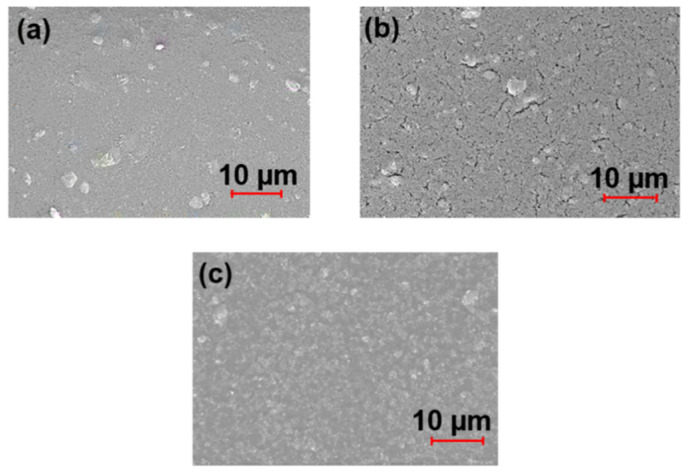
SEM images of (**a**) pure TiO_2_, (**b**) TiO_2_ – KH550 undoped with PEG, and (**c**) TiO_2_ – KH550 – PEG films.

**Figure 5 materials-15-03292-f005:**
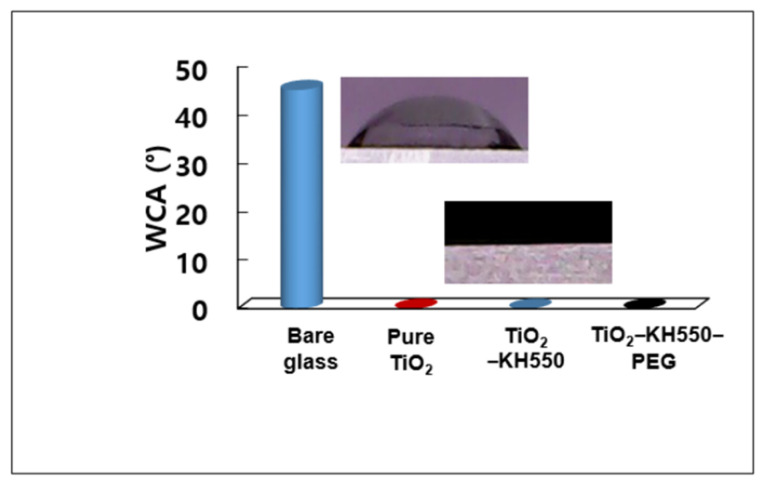
Water contact angles (WCAs) of bare glass, pure TiO_2_, TiO_2_ – KH550, and TiO_2_ – KH550 – PEG films (Insert images: Contact angles of water droplets).

**Figure 6 materials-15-03292-f006:**
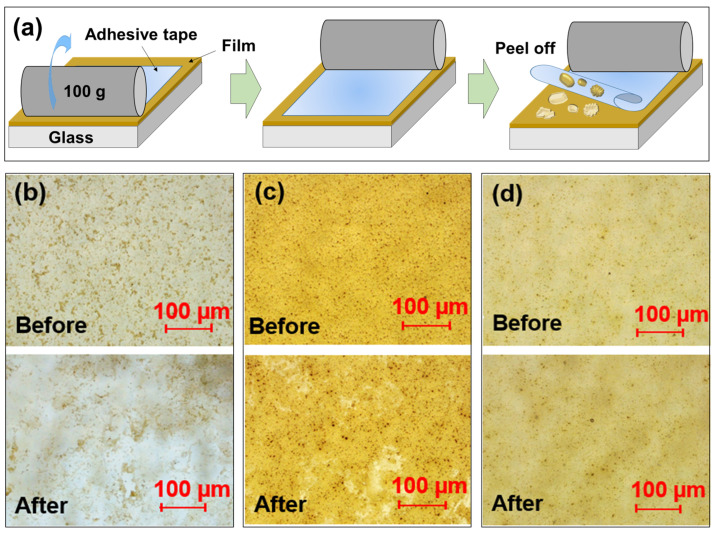
(**a**) Schematic diagram of peeling test and optical microscope images before/after peeling test on (**b**) pure TiO_2_, (**c**) TiO_2_ – KH550 and (**d**) TiO_2_ – KH550 – PEG films.

**Figure 7 materials-15-03292-f007:**
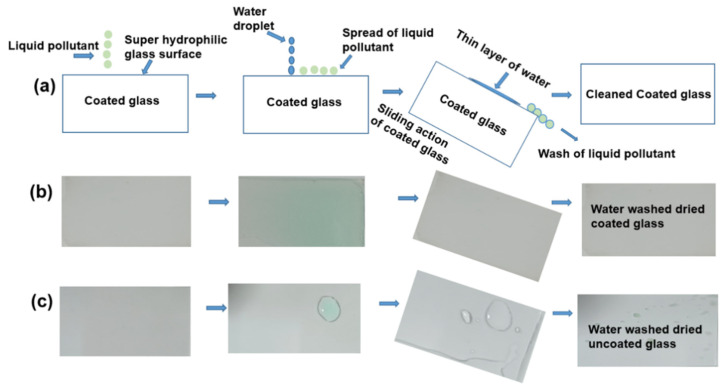
Schematic design of (**a**) self-cleaning mechanism of super-hydrophilic glass substrate, and self-cleaning behaviors of (**b**) coated glass and (**c**) bare glass.

**Figure 8 materials-15-03292-f008:**
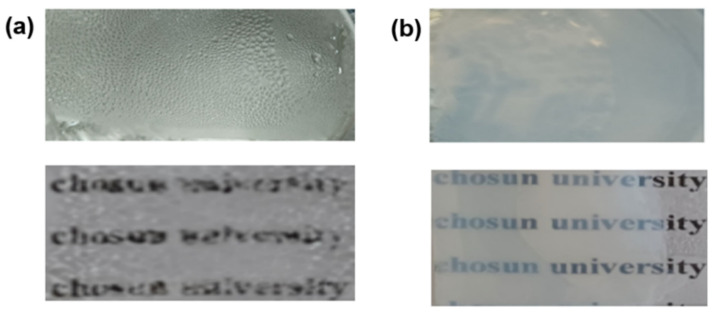
Anti-fogging properties of (**a**) bare glass and (**b**) TiO_2_ – KH550 – PEG -coated glass.

**Figure 9 materials-15-03292-f009:**
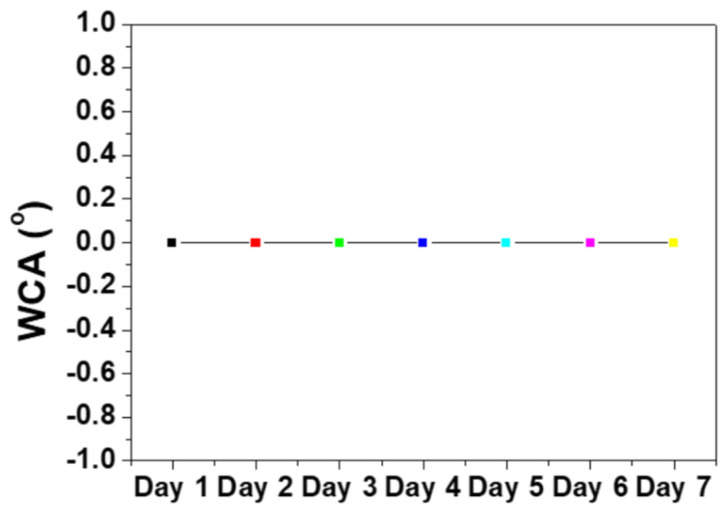
Stability of super-hydrophilicity of TiO_2_ – KH550 – PEG -coated glass in darkness.

**Figure 10 materials-15-03292-f010:**
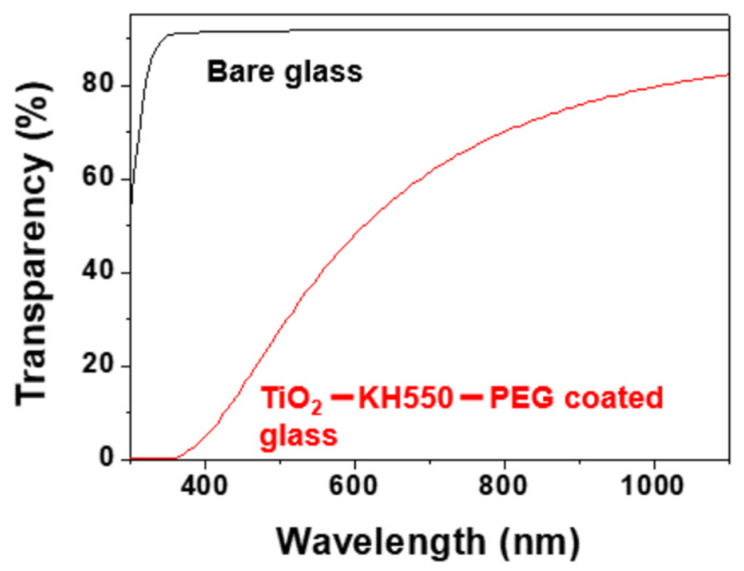
Transparency of bare glass and TiO_2_ – KH550 – PEG film on a glass substrate.

## Data Availability

Not applicable.

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
