# Peer review of "Fabrication of TiO2 ̶ KH550 ̶ PEG Super-Hydrophilic Coating on Glass Surface without UV/Plasma Treatment for Self-Cleaning and Anti-Fogging Applications"

_materials, 2022, doi:10.3390/ma15093292_

Round 1
Reviewer 1 Report
This work is devoted to the fabrication of TiO2-KH550-PEG super-hydrophilic coatings on the glass surface. The approach is interesting, but the manuscript contains some unclear points about the pure TiO2 and the PEG-removed TiO2 coatings. The author needs a fully comparative study for (1) the bare glass, (2) the pure TiO2-coated glass, (3) the PEG-removed TiO2-coated glass, and (4) the TiO2-KH550-PEG-coated glass for Figures 4, 5, 6, and 7. Additionally, the mechanism of the appearance of the super-hydrophilicity should be discussed. What is the main factor (chemical, geometrical, or any other) for the super-hydrophilicity? This referee requests a revision of the manuscript to solve the above-mentioned problems.
Author Response
We appreciate the reviewers' comments.
Please check the attached file.

Reviewer 2 Report
Dear Authors,
the preparation of a modified TiO2-layer with self-cleaning properties due to hydrophilicity is described. The coating is prepared from TiO2-nanaoparticles with PEG and the coupling agent KH550 by a single coating step, followed by a simple heat treatment at 350°C.
The focus is set, that no additional UV- or plasma-treatment is necessary. This is not really a distinguished attribute. The literature about superhydrophilic titania coatings is manifold!
What's the advantage to a pure TiO2-coating (from Ti-butoxide e.g.)?
Some measurements are not shown (UV-VIS-measurements) or the interpretation is weak (IR-transmittance). Most of the photographs are much too small and give no idea of the results.
It’s not clear, weather you use KH550 an amino-functional coupling agent, like described in the introduction, or the methacrylate (KH570?), like described in the experimental?
Hydrophilicity “also in the dark” one can find only in the introduction and in the conclusions, but no experimental data one can find in the results!
In the experimental is missed:
-preparation of the pure TiO2-films, which you use for comparision!
-kind of glass: sodalime?...silica? …?
-transmittance measurements: In Fig 7 you “show” the transparency of the samples. But how did you measure? Which machine, with or without integrating sphere?
Results:
3.2: 2852 and 2924 cm-1 show the absorption peaks of methylene-groups, yes. But you cannot know, whether they belong to PEG, residues of TEOS or KH550/70 !
What are the really big peaks at 1400 an 1700???
At 1000 cm-1 no peak is visible, but at around 1100cm-1 there is a peak, which should belong to the stretching vibration of Si-O-Si-bonds. This does not allow any conclusion to Si-O-Ti-bonds!
In ref. 35, 37, 42 one cannot find any hind that 1000cm-1 belongs to Si-O-Ti !!???
3.4 Wettability
PEG and KH550/570 include CH2-groups, which are hydrophobic. So these both materials cannot play an “significant” role in increasing hydrophilicity. The comparison to a pure TiO2-film is missed!
The photographs of the droplets in Fig.4 are much too small!
3.5 Self-cleaning test
The procedure in fig.5 is not clear, the pollutant is not visible, b) and c) are not comparable, new photographs are necessary!
3.6 (is still 3.5) Antifogging
The photographs are too small!
3.7
And what do you mean with 82% light transmittance? At a certain wavelength? Oder the average over the visible range?
You should show the transmittance spectra!
In the beginning of 3.7 you talk about 82 an 92 %, according to fig.7, but then you talk about a reduction of 8% and 70% light transmittance!?
Conclusions:
You talk about an improved adherence, but this is not tested or documented in your paper!
Good (>80%) transparency in the visible-NIR-range? UV-protective coatings?
You didn’t show any measurements to this range!
Can you give any range to the coating thickness?
Author Response

(The authors gave the same response as above.)

Reviewer 3 Report
The authors presented a method to exert superhydrophilicity to the glass surface without complicated treatments. However, the experimental design and discussion in this manuscript are not systematic and convincing. Therefore, I do not recommend its publication. Detailed comments are as follows.
- It is suggested to have a schematic illustration for the preparation of modified TiO2 film especially showing the modification states at the molecular level.
- Section 2.2.2.2: Is 350° Celsius and Fahrenheit? Why is this temperature? The authors should consider the degradation of PEG at high temperature? More details are needed in this part.
- 3.5: “The prepared colored metal solution was poured onto a coated and bare glass surface using a micropipette”? Was this surface coated or bare? Check the misunderstanding expression thorough the whole manuscript.
- Correct the wrong expression “methylene (-CH-) group”. The presence of methylene groups does not indicate the successful graft. Maybe it is the remaining of KH550. It is suggested to provide convincing evidence.
- The appearance of the coated glass and uncoated glass from Fig. 5 looks too different to get useful results. The authors should provide clearer images without complex glass reflections.
- Overall, the current investigation is not sufficient to support the advantages of the developed coating method.
Author Response

(The authors gave the same response as above.)

Round 2
Reviewer 1 Report
The scientific quality of the manuscript has been improved after the appropriate revision.
Author Response
We appreciate the reviewers' valuable comments.
Reviewer 2 Report
Dear Authors,
your manuscript looks much more complete now.
But two details still have to be clarified:
1) IR-measurement and interpretation:
You cannot be sure, that the CH2-peaks are caused by PEG!
To evaluate whether the absorption-peaks of the CH2-groups belong to PEG or KH550 (or residues of TEOS), its necessary to measure also a TiO2-KH550 film without PEG! You have to show this measurement and then you can compare and assign!
2) Transmittance data:
You show the transmittance curves now, ok!
The samples block UV, ok!
But to say, that the transmittance is reduced for only 10 % in the Vis-NIR, that's really not a precise information! This is not representative for the range you talk about! That's not a scientific way of documentation!
This value belongs only to 1100nm. In the visible range (380-780nm) the transmittance is 5-70%!
This relation has to be described more specific, in the abstract, in the results and also in the conclusions!
3) and a last point:
How did you measure film thickness? You have to explain!
Author Response
We appreciate the reviewers' valuable comments.
Please check the attached file.

Reviewer 3 Report
Accept as it is.
Author Response

(The authors gave the same response as above.)
